# Unequal access? Use of sickness absence benefits by precariously employed workers with common mental disorders: a register-based cohort study in Sweden

Julio C Hernando-Rodriguez ,[1] Nuria Matilla-Santander,[1] Chantelle Murley ,[2] Katrina Blindow,[1] Signild Kvart,[1] Melody Almroth,[1,3] Bertina Kreshpaj,[1,4] Emelie Thern,[1] Kathryn Badarin,[1] Carles Muntaner,[5,6,7] Virginia Gunn,[1,8,9] Eva Padrosa,[10,11,12] Mireia Julià,[10,11,12] Theo Bodin[1,13]

**Correspondence to**
Dr Julio C Hernando-Rodriguez; julio.hernando@ki.se

## ABSTRACT

**Objective** This study compares the use of sickness absence benefits (SABs) due to a common mental disorder (CMD) between precariously employed and non-precariously employed workers with CMDs.

**Design** Register-based cohort study.

**Participants** The study included 78215 Swedish workers aged 27–61 who experienced CMDs in 2017, indicated by a new treatment with selective serotonin reuptake inhibitors (SSRIs). Excluded were those who emigrated or immigrated, were self-employed, had an annual employment-based income <100 Swedish Krona, had >90 days of unemployment per year, had student status, had SABs due to CMDs during the exposure measurement (2016) and the two previous years, had an SSRI prescription 1 year or less before the start of the SSRI prescription in 2017, had packs of >100 pills of SSRI medication, had a disability pension before 2017, were not entitled to SABs due to CMDs in 2016, and had no information about the exposure.

**Outcome** The first incidence of SABs due to CMDs in 2017.

**Results** The use of SABs due to a CMD was slightly lower among precariously employed workers compared with those in standard employment (adjusted OR [aOR] 0.92, 95% CI 0.81 to 1.05). Particularly, women with three consecutive years in precarious employment had reduced SABs use (aOR 0.48, 95% CI 0.26 to 0.89), while men in precarious employment showed weaker evidence of association. Those in standard employment with high income also showed a lower use of SABs (aOR 0.74, 95% CI 0.67 to 0.81). Low unionisation and both low and high-income levels were associated with lower use of SABs, particularly among women.

**Conclusions** The study indicates that workers with CMDs in precarious employment may use SABs to a lower extent. Accordingly, there is a need for (1) guaranteeing access to SABs for people in precarious employment and/or (2) reducing involuntary forms of presenteeism.

## BACKGROUND

In the past decades, many countries, including Sweden, have experienced changes in labour market policies that prioritise organisational flexibility over employment security.[1] This led to an increase in precarious employment (PE),[2] which is characterised by employment insecurity, insufficient income and a lack of rights and social protection.[3 4] The impact of these labour market changes on the health of the workforce and the use of sickness absence benefits (SABs) is not fully understood, especially, among workers in PE.

Mental disorders are one of the major causes of the global burden of disease.[5] In Sweden, half of the working-age population experiences a mental health condition during their lifetime; every year, 1 million workers have a mental disorder.[6] As a result, mental disorders are one of the leading diagnoses causing sickness absence (SA), especially among young people.[7] SA rates are not only driven by the health and work capacity of the working population but are influenced by contextual factors. Therefore, it has been suggested that low rates of SA in insecure jobs may reflect higher levels of presenteeism (ie, working while being ill) due to workers' fear of dismissal rather than better health status.[8 9]

## STRENGTHS AND LIMITATIONS OF THIS STUDY

⇒ This study uses population registers with high validity and low attrition rates.
⇒ The linkage of work disability and pharmaceutical registers enables accurate measurement of sickness absence in individuals in formal employment receiving new common mental disorders treatment.
⇒ The longitudinal design of this study minimises the potential for reverse causation.
⇒ Potential underestimation of findings by requiring 3 years of exposure to precarious employment (those with longer exposure tend to be healthier) is mitigated in the 1-year analysis.

In Sweden, the proportion of the working-age population who are at risk of not being entitled to SA is among the lowest (less than 1%) in the European Union.[10] Those who are not entitled to SA are often family workers employed in small family businesses and who do not receive a formal salary and, therefore, do not pay social contributions.[10] However, in 2016, the Swedish Social Insurance Inspectorate (ISF) deemed temporary employees' access to SABs to be legally uncertain, mainly due to the design of regulations that leave room for interpretation.[11] Workers without fixed schedules are particularly affected because this causes ambiguity as to what hours they would have worked had they not been sick, which opens the possibility to consider non-working periods as unemployment. The regulations of SABs are different for employees and unemployed persons, including entitlement requirements and the level of compensation. Moreover, the ISF found that temporary employees had fewer days of SABs than those permanently employed.

Previous studies have examined demographic characteristics such as age and gender, singular employment characteristics, trade union membership or work environment factors in relation to SA.[12–15] Some of these determinants of SA have been examined specifically among people with common mental disorders (CMDs—stress-related disorders, depression and anxiety).[16] To our knowledge, only two previous cross-sectional studies have analysed the relationship between PE and SA, and both of which found a positive association.[17 18] This is in line with previous studies confirming a relationship between PE and poor health, including a systematic review of longitudinal studies.[19] A recent Swedish study using the same data sources as the present study found that trajectories with constant PE increased the risk of CMDs.[20]

Accordingly, we hypothesise that workers in PE would have a higher likelihood of using SABs due to CMDs when compared with those in standard employment due to their worse mental health status. However, workers in PE who formally qualify for SABs may still experience barriers to accessing their rights. To investigate the use of SABs, we include only individuals who have recently initiated treatment for CMDs and who formally qualify for SABs.

This study aims to compare the use of SABs due to CMDs between precariously employed and non-precariously employed workers with CMDs.

## METHODS

### Study population

This register-based study is based on the Swedish Work, Illness and Labour-market Participation (SWIP) cohort of the total Swedish population. The SWIP cohort is obtained through the linkage of multiple registers and includes all individuals aged 16–65 years who were registered in Sweden in 2005 (around 5.7 million individuals), with follow-up until 2017. This study uses a subpopulation of 78 215 individuals aged 27–61 years in 2016 who

experienced a CMD in 2017, indicated by the initiation of treatment with selective serotonin reuptake inhibitors (SSRIs). Included individuals had similar characteristics to the Swedish workforce, except for the higher share of women (online supplemental file 1).

Sociodemographic and employment data were obtained from the Longitudinal Integration Database for Health Insurance and Labour Market Studies for 2016. Data on SA (from day 15 onwards) were taken from the Micro Data for Analysis of Social Insurance (MiDAS) registers and included information such as the International Classification of Diseases tenth edition (ICD-10) diagnosis and length of SA for 2017. The Swedish Prescribed Drug Register informed the prescribed medicines dispensed from pharmacies for 2017.

Exclusion criteria were as follows: (1) emigration or immigration (2012–2016), (2) self-employment during 2014–2017 because SABs regulations differ,[21] (3) annual employment-based income < 100 Swedish Krona (SEK) to ensure working status, (4) >90 days of unemployment per year to avoid measuring the effect of unemployment on the outcome, (5) student status during the autumn term to avoid misclassification as PE, (6) SA due to CMDs during the exposure measurement (2016) and the two previous years (2014–2015) to reduce the risk of reverse causation, (7) SSRI prescription 1 year or less before the start of SSRI prescription in 2017,[22 23] (8) packs of >100 pills of SSRI medication in 2017, as this is uncommon for new SSRI treatment, (9) disability pension before 2017, (10) no entitlement to SABs due to CMDs in 2016 and (11) incomplete information on the exposure variable in 2016 (2014–2016 in the sensitivity analysis) (figure 1).

### Exposure variable

The PE measure was based on the second version of the Swedish Register-based Operationalisation of Precarious Employment (SWE-ROPE).[24] The SWE-ROPE considers five items (ie, contractual relationship insecurity, contractual temporariness, multiple jobs/economic sectors, income level and lack of unionisation) that represent the three dimensions of PE (ie, employment insecurity, income inadequacy and lack of rights and protection). For each individual, an annual score for each of the five items was calculated.[25] Scores ranged by item: contractual employment insecurity (agency employed and directly employed: from −1 and 0); contractual temporariness (unstable employment and stable employment: −2 and 0); multiple jobs/economic sectors (multiple jobs and sectors, multiple jobs, a single job: −2 to 0), income level (<60% of the median, 60%–79%, 80%–119%, 120%–200% and >200%: −2 to+2) and lack of unionisation (<70% of collective bargaining agreement coverage, 70%–90% and >90%: −2 to 0). The median income (SEK375 300 in 2016) was based on the Swedish working population after applying the inclusion and some of the main exclusion criteria (1–5 and 9 described above).

A final summative score was obtained by summing the total score of the five items. Lower final scores suggest PE,

Figure 1. Flow chart of the total study population.

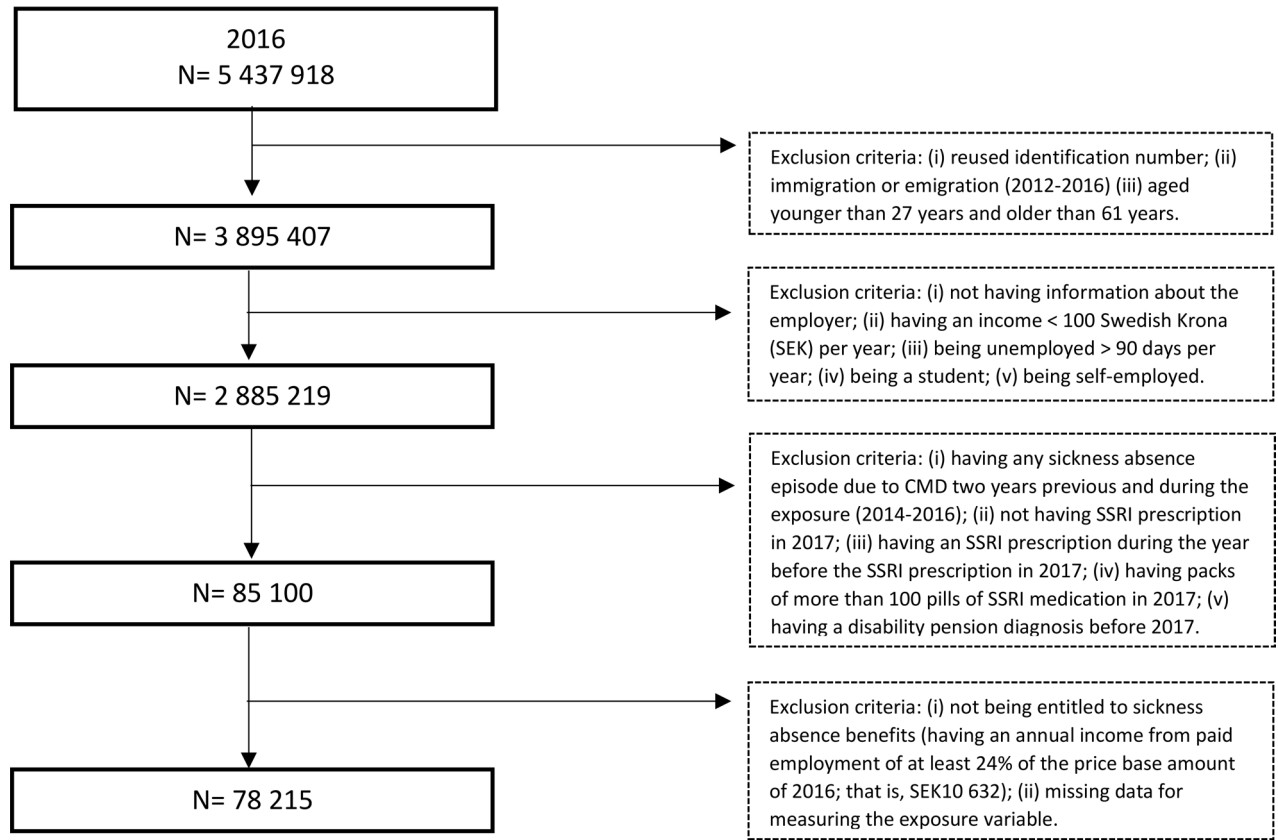

**Figure 1** Flow chart of the total study population. This is a flow chart of the exclusion criteria applied (there is no legend). CMD, common mental disorder; SSRI, selective serotonin reuptake inhibitor.

while a higher score suggests non-PE. The final summative score for 2016 was categorised into four employment quality typologies: PE relationship (PER; score <−3), substandard employment relationship (SSER; a score between −3 and −1), standard employment relationship (SER; score=0) and SER with high income (SER/HI; score>0). The SER/HI group was defined in light of previous literature[20 26] and included more than 85% of individuals with HI levels (120%–199% of the median income).

### Outcome variable

The outcome was the first occurrence of SABs due to CMDs starting in the 2 months spanning the month before and after the SSRI prescription date in 2017 (categorical outcome: 0, having SSRI and no SABs; 1, having SSRI co-occurring with SABs). This period was selected based on literature observing the effect of SSRI treatment after 3–4 weeks, so the risk of experiencing a new SA spell after that time may decrease.[27–29] The SSRI prescriptions included antidepressants from N06AB02 to N06AB10 according to the Anatomical Therapeutic Chemical classification system. SA due to CMDs was identified by ICD-10 codes: depression (F32–F33), anxiety (F41) and stress-related disorders (F43) including exhaustion disorder, which is conceptually similar to burnout. This definition of CMDs is based

on a previous paper about PE and the risk of CMDs.[20] Regarding the SABs, all people living in Sweden aged 16 and over who receive a minimum annual income from work of at least 24% of the price base amount (PBA) and who have a reduced work capacity of at least 25% of their usual working hours due to disease or injury are covered by the SA insurance system.[30] In this study, SEK10 632 (0.24*PBA=0.24*44300 in 2016) was used to estimate eligibility for SABs outcome in 2017. A physician's certificate is required after the first 7 days of SA. SABs replace up to 80% of lost work income. After the first qualifying day without compensation, the employer pays SA for the first 13 days. Therefore, SA spells of less than 15 days are not registered in MiDAS. From the 15th day onwards, SABs are paid by the Social Insurance Agency. For the present study, SABs due to CMDs included only SA spells of at least 15 days.

### Potential confounding variables

At baseline, age, education level, country of birth, family composition (civil status and having children), unemployment duration, economic sector, area of residence, occupation and comorbidity (2016). Comorbidity was measured as having SA due to any diagnoses other than CMDs, and it is known that SA episodes in

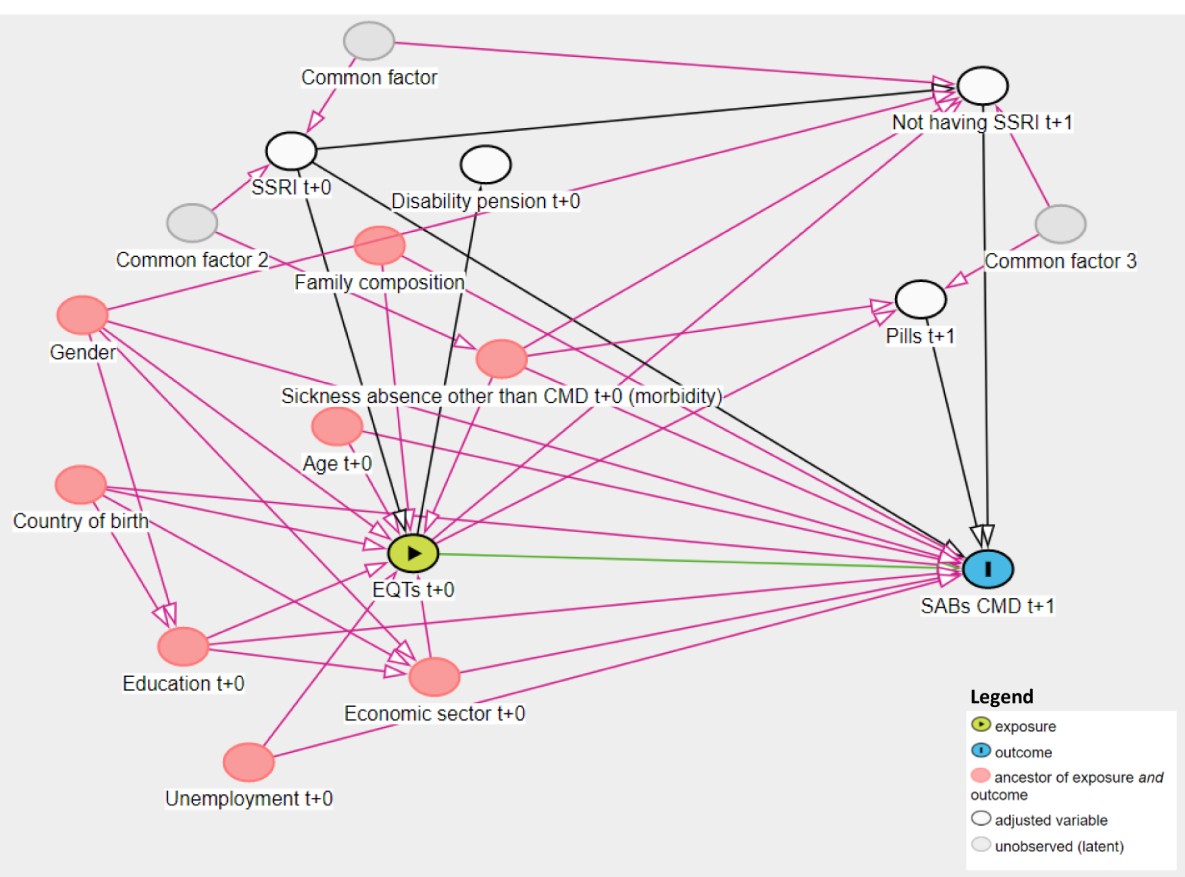

**Figure 2** Directed acyclic graph of the relation between employment quality typologies (EQT) and sickness absence benefits (SABs) due to common mental disorders (CMDs). Minimal sufficient adjustment sets for estimating the direct effect of EQTs t+0 (baseline, 2016) on SABs due to CMD t+1 is age, gender, education, country of birth, family composition, economic sector, unemployment and sickness absence due to diagnoses other than CMD, having selective serotonin reuptake inhibitors (SSRI) prescription during the year before the SSRI prescription in 2017, having disability pension before 2017, having packs of more than 100 pills in 2017 and not having SSRI prescriptions close to the SABs in 2017.

the previous years showed to be a predictor for future episodes of SA.[31]

### Statistical analysis

Logistic regression models were used to estimate the association between employment quality typologies and the use of SABs due to CMDs in 2017. A sensitivity analysis was conducted by conditioning models to have the same exposure for three consecutive years (2014–2016) to observe if stability could influence the results. Models were adjusted for the potential confounders mentioned above and identified using a directed acyclic graph (figure 2). All analyses were conducted for the total population as well as women and men separately. The analysis is separated by gender because previous studies have found different patterns in SA among women and men.[32 33] Women have higher SA rates compared with men possibly due to the higher exposure to an unequal division of domestic work combined with paid work.[32] Work–family conflicts are related to higher ORs to report short SA (1–7 days) among men and longer SA among women (8–30 days).[33] Interaction effects of gender and the main exposure were tested. We would like to

clarify our approach to interpreting the results of the models by considering CIs as 'compatibility intervals' as recommended by Amrhein *et al*.[34] Rather than solely focusing on statistical significance, we examined the effect sizes that are most compatible with our data.

Data management and statistical analysis were performed using Stata SE V.17.

### Patient and public involvement

Patients and/or the public were not involved in the design, or conduct, or reporting, or dissemination plans of this research.

### RESULTS

People in PER (5.10%) in general had a higher proportion of young-middle aged individuals (27–40 years) (59.4%), women (69.7%), low educational attainment (63.2%), born in Sweden (88.4%), couples with children (44.4%), zero days unemployed (85.6%), private sector (79.6%), employed by an agency (5.5%), unstable employment (93.5%), multiple job holding in different economic sectors (20.7%), income level below 60% of the median (47.5%) and low unionisation (22.6%) compared

with the other employment quality typologies (table 1). The proportion of people having a first spell of SABs due to CMDs in 2017 was 7.5% among our study population and showed differences according to sociodemographic and employment characteristics in general (online supplemental table 2), and by gender and employment quality typology (online supplemental tables 3 and 4, respectively).

The association between 1-year exposure to employment quality typologies and the use of SABs due to CMDs exhibited an inverted U-shape pattern, with lower adjusted OR (aOR) estimates found among individuals with an SER/HI (aOR 0.74, 95% CI 0.67 to 0.81), followed by near-significance estimates of those in a PER (aOR 0.92, 95% CI 0.81 to 1.05), compared with the SER group (table 2). This association between PER and SABs suggests 8% reduced risk. However, our data are also compatible with a range of risk differences, spanning from a reduction of 19% to a small increase of 5%. The SSER group showed no significant differences compared with the SER group. Women were less likely to use SABs due to CMDs in the PER (OR 0.88, 95% CI 0.75 to 1.04) and the SER/HI groups (aOR 0.72, 95% CI 0.63 to 0.83) compared with the SER group. Men in SSER and PER showed weaker evidence of association, except for those in the SER/HI group (aOR 0.76, 95% CI 0.66 to 0.86). Overall, regression models conditional on having the same exposure for three consecutive years showed similar results, in this case, lower use of SABs due to CMDs occurred among individuals with SER/HI (aOR 0.74, 95% CI 0.65 to 0.85). Women with three consecutive years of PER had the lowest probability of using SABs due to CMDs (aOR 0.48, 95% CI 0.26 to 0.89), followed by women and men with SER/HI compared with the SER group (aOR 0.76, 95% CI 0.62 to 0.92 and aOR 0.73, 95% CI 0.61 to 0.88, respectively). Given that area of residence and occupation barely changed the estimates, they were not included in the models as confounders.

We tested interaction effects between gender and employment quality. Women in PE had a 27% lower risk of SABs compared with men in standard employment after controlling for the same potential confounding factors (data not shown). The effect of the interaction term was somewhat pronounced for women in PER, but, in general, differences between gender and employment quality categories were relatively small. However, when comparing the model without the interaction term to the model containing it, the likelihood ratio test was not statistically significant (p>0.3955) (data not shown). This suggests that including the interaction term does not improve the fit of the model. Therefore, we have kept the original model without the interaction term.

The employment quality items most strongly related to the use of SABs due to CMDs were income and unionisation levels (table 3). In the fully adjusted model, HI was associated with a reduced probability of SABs due to CMDs (120%–200% of the median income—aOR 0.79, 95% CI 0.72 to 0.85; and more than 200% of the

median income—aOR 0.61, 95% CI 0.49 to 0.75), while the lowest-income level showed slight differences (lower than 60% of the median income—aOR: 0.92, 95% CI 0.82 to 1.03). Women with the highest-income levels exhibit lower aORs compared with men (aOR 0.48, 95% CI 0.33 to 0.71 and aOR 0.71, 95% CI 0.54 to 0.92, respectively). Low unionisation coverage was associated with lower use of SABs (less than 70% of unionisation level—aOR 0.82, 95% CI 0.68 to 0.99), particularly among women (aOR 0.77, 95% CI 0.58 to 1.03, near to significance). Similar results were found for women and men with medium unionisation coverage. The unadjusted ORs and two different model adjustments can be observed in online supplemental table 5.

## DISCUSSION

This study indicated that having a PER was associated with a lower probability of using SABs due to CMDs compared with SER, especially among women. On the other side of the spectrum, we also found that SER/HI was strongly and negatively associated with SABs. The income and unionisation level were the themes of employment quality most strongly related to using SABs due to CMDs: having HI and low unionisation levels were associated with lower use of SABs due to CMDs, particularly, among women.

Our results on the negative association between PER and the use of SABs due to CMDs contrast with previous studies that found an increased prevalence of self-reported SA due to health problems among precariously employed workers, in Sweden and other countries.[17 18] In our present study, sickness presenteeism (SP) may explain the lower odds of SABs due to CMDs in the PER group. Precariously employed workers who typically have low-income jobs are likely to incur SP because they cannot afford to use SABs. Using SABs results in a period of reduced income since the SABs only partially replace wages.[35] Additionally, SP may stem from the fear of being dismissed due to employment insecurity or from the lack of union protection.[9 36] An earlier study found that workers who transitioned from temporary to permanent employment had an increased probability of SA.[8] Precariously employed workers in our study population had the lowest unionisation levels and low unionisation was related to lower use of SABs due to CMDs. These findings are consistent with previous studies in Germany and Norway where trade union members had higher SA than non-members.[14 15] Furthermore, SP can be partially explained by the social stigma surrounding mental health, which may originate from colleagues and employers.[37] This stigma can lead individuals to feel more validated in taking SA when their illness is physical, rather than when it is related to mental health. A second possible explanation for the low use of SABs among precariously employed workers is related to legal uncertainty when the Social Insurance Agency assesses entitlement to SABs for non-standard employees.[38] If temporary employees have scheduled work, they receive SABs for the days that they would have worked had the illness not reduced their

**Table 1**  Sociodemographic and employment characteristics at baseline (2016) in a cohort of 78 215 individuals with common mental disorders in 2017, aged 27–61, entitled to sickness absence benefits in 2016 by employment quality typology (baseline, 2016)

| | SER/HI (14.68%) | SER (38.04%) | SSER (42.18%) | PER (5.10%) |
| --- | --- | --- | --- | --- |
| | N (%) | N (%) | N (%) | N (%) |
| Age | | | | |
| 27–40 years | 2878 (25.1) | 9817 (33.0) | 15 612 (47.3) | 2368 (59.4) |
| 41–51 years | 5239 (45.6) | 11 432 (38.4) | 10 844 (32.9) | 1026 (25.7) |
| 52–61 years | 3366 (29.3) | 8504 (28.6) | 6537 (19.8) | 592 (14.9) |
| Gender | | | | |
| Male | 5999 (52.2) | 10 133 (34.1) | 8554 (25.9) | 1208 (30.3) |
| Female | 5484 (47.8) | 19 620 (65.9) | 24 439 (74.1) | 2778 (69.7) |
| Education level | | | | |
| Elementary/high school education | 3395 (29.6) | 15 831 (53.2) | 19 329 (58.6) | 2517 (63.2) |
| Higher education <3 years | 1989 (17.3) | 4384 (14.8) | 4446 (13.5) | 637 (16.0) |
| Higher education ≥3 years | 6092 (53.1) | 9528 (32.0) | 9201 (27.9) | 830 (20.8) |
| Country of birth | | | | |
| Sweden | 10 449 (91.0) | 26 561 (89.3) | 29 203 (88.6) | 3523 (88.4) |
| Not Sweden | 1031 (9.0) | 3174 (10.7) | 3770 (11.4) | 463 (11.6) |
| Family composition | | | | |
| Couple with children | 6045 (52.6) | 13 320 (44.8) | 16 338 (49.5) | 1768 (44.4) |
| Couple without children | 1354 (11.8) | 3630 (12.2) | 3151 (9.6) | 286 (7.2) |
| Single with children | 1252 (10.9) | 3903 (13.1) | 4528 (13.7) | 560 (14.0) |
| Single without children | 2832 (24.7) | 8900 (29.9) | 8976 (27.2) | 1372 (34.4) |
| Unemployment | | | | |
| 0 days unemployed | 11 429 (99.5) | 29 514 (99.2) | 31 733 (96.2) | 3414 (85.6) |
| From 1 to 90 days unemployed | 54 (0.5) | 239 (0.8) | 1260 (3.8) | 572 (14.4) |
| Economic sector | | | | |
| Private | 6995 (60.9) | 13 908 (46.7) | 17 578 (53.3) | 3172 (79.6) |
| Public | 4488 (39.1) | 15 845 (53.3) | 15 415 (46.7) | 814 (20.4) |
| Contractual relationship insecurity | | | | |
| Directly employed by an employer | 11 473 (99.9) | 29 715 (99.9) | 32 529 (98.6) | 3765 (94.5) |
| Employed by an agency | 10 (0.1) | 38 (0.1) | 464 (1.4) | 221 (5.5) |
| Contractual temporariness | | | | |
| Stable employment | 11 483 (100.0) | 29 282 (98.4) | 18 070 (54.8) | 260 (6.5) |
| Unstable employment | (0.0) | 471 (1.6) | 14 923 (45.2) | 3726 (93.5) |
| Multiple jobs/economic sectors | | | | |
| 1–2 jobs | 11 143 (97.1) | 29 123 (97.9) | 30 584 (92.7) | 2168 (54.4) |
| 3 or more jobs | 221 (1.9) | 440 (1.5) | 1916 (5.8) | 994 (24.9) |
| 3 or more jobs in three or more sectors | 119 (1.0) | 190 (0.6) | 493 (1.5) | 824 (20.7) |
| Income level* | | | | |
| <60% of the median | (0.0) | (0.0) | 3320 (10.1) | 1892 (47.5) |
| 60%–79% of the median | (0.0) | (0.0) | 14 823 (44.9) | 1153 (28.9) |
| 80%–119% of the median | (0.0) | 28 657 (96.3) | 10 708 (32.5) | 878 (22.0) |
| 120%–200% of the median | 9849 (85.8) | 590 (2.0) | 3960 (12.0) | 62 (1.5) |
| >200% of the median | 1634 (14.2) | 506 (1.7) | 182 (0.5) | 1 (<0.1) |
| Unionisation level | | | | |
| <70% | (0.0) | 20 (0.1) | 852 (2.6) | 901 (22.6) |

Continued

**Table 1** Continued

| | SER/HI (14.68%) | SER (38.04%) | SSER (42.18%) | PER (5.10%) |
| | N (%) | N (%) | N (%) | N (%) |
|---|---|---|---|---|
| 70%–90% | 55 (0.5) | 412 (1.4) | 3297 (10.0) | 866 (21.7) |
| >90% | 11 428 (99.5) | 29 321 (98.5) | 28 844 (87.4) | 2219 (55.7) |

Missing values in the educational level (SER/HI: 7, 0.06%; SER: 10, 0.03%; SSER: 17, 0.05%; PE: 2, 0.05%) and country of birth (SER/HI: 3, 0.03%; SER: 18, 0.06%; SSER: 20, 0.06). Persons entitled to sickness absence benefits in 2016 are those having an annual income from paid employment of at least 24% of the price base amount of 2016; that is, SEK10 632.
*Income level categories based on the median of the Swedish working population (SEK375 300K) after applying several inclusion and exclusion criteria (see the Methods section). Income category boundaries: <60% of the median (SEK225 180), 60%–79% of the median (SEK225 180–SEK296 487K), 80%–119% of the median (SEK300 240–SEK446 607 SEK), 120%–200% of the median (SEK450 360–SEK746 847), and >200% of the median (SEK750 600).
PE, precarious employment; PER, precarious employment relationship; SER, standard employment relationship; SER/HI, SER with high income; SSER, sub-SER.

work capacity by at least 25%. Non-standard employees, however, often lack a formal schedule covering the days in question, for example, on-demand employees get their shifts assigned on a day-to-day basis or have a schedule based on an oral agreement, and risk being considered unemployed in the event of sickness and cannot access SABs from the employer. To access SABs, work capacity must be assessed, and this is done differently according to employment status. The work capacity of employees is assessed only relative to their current employment for the first 90 days of SA and only later against all jobs in the entire labour market, but unemployed (or non-standard workers without scheduled hours) will be assessed relative to all jobs in the entire labour market from the start. In practice, this often means that they do not qualify for SABs, since other jobs may exist in the labour market where their current functional ability does not impose a sufficient reduction in work capacity. However, we may bear in mind that this explanation is less plausible for workers with CMDs that may have a reduced work capacity for any task.

In general, women had lower use of SABs in any employment quality typology compared with the SER, and the association was more pronounced in women than in men. These results were unexpected since women usually have higher SA rates than men.[39] However, our study population only included individuals with SSRI treatment (two-thirds of them women), which has not been studied before with this study design.[40] The lower use of SABs among women is plausible because it has been suggested that women may feel a greater responsibility to their colleagues and avoid burdening them by cooperating more when performing work tasks compared with men.[35] Furthermore, another study showed that female-dominated jobs related to jobs with responsibility for others such as education, human health and social work activities are more prone to SP.[41] However, this explanation is unlikely to apply to our study since we accounted for the potential effect of women's concentration in certain sectors on SABs by controlling for the sector in the analysis. Women with a consecutive 3-year period in PER had the lowest use of SABs. This

could be explained by the fact that women are more frequently employed in temporary employment, which has been related to fewer days of SABs than permanently employed and, probably, to lower access to SABs for the reasons previously mentioned.[38]

SER/HI showed the highest income levels and we found that having a HI was related to a lower likelihood of using SABs. In this line, a Finnish study found that a higher position by education, occupational class and income was related to lower SA.[42] Another possible explanation for the lower use of SABs in employees in SER/HI is also SP, but for different reasons than the PER group. In a study conducted in Norway and Sweden, having supervisory responsibilities at work that no one else can perform was an influential factor for SP, which could be the case for the high earners in the present study.[35] Another driver for SP among high earners could be work satisfaction since they probably have jobs with more creativity, freedom and control.[35 43] Another explanation is that high earners have more knowledge-intensive occupations where they can work from home rather than take SA.[44] Lastly, individuals in SER/HI could plausibly be more likely to combine SSRIs with cognitive–behavioural therapy (CBT) or mindfulness which could be, to a certain extent, protective of work capacity.[45 46]

SA can also be conceptualised as a health status measure. Accordingly, an important consequence of SP is an increased risk of more severe and long-lasting health problems that, in turn, lead to longer periods of SA, regardless of the employment quality typology followed.[47]

### Limitations and strengths
Limitations of this study could arise from the design of the study when analysing individuals with 3 years with the same exposure since those with PE are healthier enough to not exit the labour market, which could underestimate the results. However, we also analysed 1 year of exposure before SABs, finding comparable results which may overcome the previous limitation. Another limitation is the possibility of residual confounding from unobserved factors when inferring the impact of employment quality

**Table 2** Association between employment quality typologies (baseline, 2016) and sickness absence benefits due to common mental disorders in 2017 among people entitled to sickness absence benefits in 2016 (N=78215)

| | N | Cases N (%) | OR (95% CI) | aOR (95% CI) | Models with 3 years of continuous exposure* (N=38 761) | | | |
| --- | --- | --- | --- | --- | --- | --- | --- | --- |
| | | | | | N | Cases N (%) | OR (95% CI) | aOR (95% CI) |
| **Employment quality typology** | | | | | | | | |
| SER | 29753 | 2268 (7.6) | 1 | 1 | 17353 | 1207 (7.0) | 1 | 1 |
| SER/HI | 11483 | 593 (5.2) | 0.66 (0.60 to 0.72) | 0.74 (0.67 to 0.81) | 6924 | 330 (4.8) | 0.67 (0.59 to 0.76) | 0.74 (0.65 to 0.85) |
| SSER | 32993 | 2667 (8.1) | 1.07 (1.01 to 1.13) | 0.99 (0.93 to 1.05) | 14065 | 1055 (7.5) | 1.08 (1.00 to 1.18) | 1.01 (0.92 to 1.10) |
| PER | 3986 | 320 (8.0) | 1.06 (0.94 to 1.20) | 0.92 (0.81 to 1.05) | 419 | 24 (5.7) | 0.81 (0.54 to 1.23) | 0.74 (0.48 to 1.12) |
| **Women (N=52 321)** | | | | | | | | |
| SER | 19620 | 1375 (7.0) | 1 | 1 | 11137 | 715 (6.4) | 1 | 1 |
| SER/HI | 5484 | 247 (4.5) | 0.63 (0.54 to 0.72) | 0.72 (0.63 to 0.83) | 2954 | 129 (4.4) | 0.67 (0.55 to 0.81) | 0.76 (0.62 to 0.92) |
| SSER | 24439 | 1892 (7.7) | 1.11 (1.04 to 1.20) | 1.00 (0.93 to 1.08) | 11138 | 810 (7.3) | 1.14 (1.03 to 1.27) | 1.00 (0.90 to 1.12) |
| PER | 2778 | 198 (7.1) | 1.02 (0.87 to 1.19) | 0.88 (0.75 to 1.04) | 296 | 11 (3.7) | 0.56 (0.31 to 1.03) | 0.48 (0.26 to 0.89) |
| **Men (N=25 894)** | | | | | | | | |
| SER | 10133 | 893 (8.8) | 1 | 1 | 6216 | 492 (7.9) | 1 | 1 |
| SER/HI | 5999 | 346 (5.8) | 0.63 (0.56 to 0.72) | 0.76 (0.66 to 0.86) | 3970 | 201 (5.1) | 0.62 (0.52 to 0.73) | 0.73 (0.61 to 0.88) |
| SSER | 8554 | 775 (9.1) | 1.03 (0.93 to 1.14) | 0.97 (0.88 to 1.08) | 2927 | 245 (8.4) | 1.06 (0.91 to 1.25) | 1.01 (0.86 to 1.19) |
| PER | 1208 | 122 (10.1) | 1.16 (0.95 to 1.42) | 1.02 (0.83 to 1.25) | 123 | 13 (10.6) | 1.37 (0.77 to 2.46) | 1.31 (0.73 to 2.36) |

OR: unadjusted OR; aOR: OR adjusted by age, gender, education, country of birth, family composition, economic sector, unemployment, and SA due to diagnoses other than CMD in the baseline (2016) Note: Persons entitled to sickness absence benefits are those having an annual income from paid employment of at least 24% of the price base amount of 2016; that is, SEK10632. Area of residence and occupation barely changed the estimates; therefore, they were not included in the models as confounders.

*Models conditioned to have the same exposure for three consecutive years (2014–2016).

aOR, adjusted OR; CMD, common mental disorder; PER, precarious employment relationship; SA, sickness absence; SER, standard employment relationship; SER/HI, SER/high income; SSER, sub-SER.

**Table 3** Association between employment quality dimensions (2016, baseline) and sickness absence benefits due to common mental disorders in 2017 (N=78215) among people entitled to sickness absence benefits in 2016

| | | | Women (N=52321) | | Men (N=25894) | |
|---|---|---|---|---|---|---|
| | Cases N (%) | aOR (95% CI) | Cases N (%) | aOR (95% CI) | Cases N (%) | aOR (95% CI) |
| **Contractual relationship insecurity** | | | | | | |
| Directly employed by employer | 5788 (7.5) | 1 | 3683 (7.1) | 1 | 2105 (8.2) | 1 |
| Employed by an agency | 60 (8.2) | 0.96 (0.73 to 1.25) | 29 (6.5) | 0.85 (0.58 to 1.24) | 31 (10.9) | 1.08 (0.74 to 1.58) |
| **Contractual temporariness** | | | | | | |
| Stable employment | 4351 (7.4) | 1 | 2792 (7.1) | 1 | 1559 (8.0) | 1 |
| Unstable employment | 1497 (7.8) | 1.02 (0.95 to 1.09) | 920 (7.2) | 1.01 (0.92 to 1.10) | 577 (9.0) | 1.04 (0.93 to 1.16) |
| **Multiple jobs/economic sectors** | | | | | | |
| 1–2 jobs | 5436 (7.4) | 1 | 3463 (7.1) | 1 | 1973 (8.2) | 1 |
| 3 or more jobs | 279 (7.8) | 1.05 (0.93 to 1.20) | 175 (7.0) | 1.00 (0.85 to 1.18) | 104 (9.6) | 1.18 (0.95 to 1.46) |
| 3 or more jobs in 3 or more sectors | 133 (8.2) | 1.09 (0.91 to 1.31) | 74 (7.3) | 1.04 (0.81 to 1.32) | 59 (9.6) | 1.19 (0.90 to 1.57) |
| **Income level*** | | | | | | |
| 80%–119% of the median | 3113 (7.7) | 1 | 1876 (7.1) | 1 | 1237 (8.9) | 1 |
| <60% of the median | 395 (7.6) | 0.92 (0.82 to 1.03) | 319 (7.3) | 0.91 (0.81 to 1.04) | 76 (8.9) | 0.94 (0.73 to 1.20) |
| 60%–79% of the median | 1415 (8.9) | 1.10 (1.03 to 1.18) | 1111 (8.4) | 1.08 (1.00 to 1.17) | 304 (10.9) | 1.16 (1.01 to 1.33) |
| 120%–200% of the median | 831 (5.8) | 0.79 (0.72 to 0.85) | 378 (5.1) | 0.79 (0.70 to 0.88) | 453 (6.4) | 0.80 (0.71 to 0.89) |
| >200% of the median | 94 (4.1) | 0.61 (0.49 to 0.75) | 28 (2.9) | 0.48 (0.33 to 0.71) | 66 (4.9) | 0.71 (0.54 to 0.92) |
| **Unionisation level** | | | | | | |
| >90% | 5409 (7.5) | 1 | 3497 (7.2) | 1 | 1912 (8.3) | 1 |
| 70%–90% | 315 (6.8) | 0.82 (0.73 to 0.92) | 161 (6.1) | 0.83 (0.70 to 0.98) | 154 (7.7) | 0.80 (0.67 to 0.96) |
| <70% | 124 (7.0) | 0.82 (0.68 to 0.99) | 54 (5.8) | 0.77 (0.58 to 1.03) | 70 (8.2) | 0.84 (0.65 to 1.09) |

aOR: OR adjusted mutually by the other employment quality dimensions and age, gender, education, country of birth, family composition, economic sector, unemployment and SA due to diagnoses other than CMD in the baseline (2016). Note: Persons entitled to sickness absence benefits are those having an annual income from paid employment of at least 24% of the price base amount of 2016; that is, SEK10 632.

*Income level categories based on the median of the Swedish working population (SEK375 300) after applying several inclusion and exclusion criteria (see the Methods section). Income category boundaries: <60% of the median (SEK225 180), 60%–79% of the median (SEK225 180–SEK296 487), 80%–119% of the median (SEK300 240–SEK446 607), 120%–200% of the median (SEK450 360–SEK746 847) and >200% of the median (SEK750 600). Area of residence and occupation barely changed the estimates; therefore, they were not included in the models as confounders.

aOR, adjusted OR; CMD, common mental disorder; SA, sickness absence.

on SABs. We cannot rule out that some people in the study population were not entitled to SABs since apart from the prior income requirement, it is necessary to have a reduced work capacity due to the CMD of at least 25%, which is not possible to assess in the datasets. Selection bias could be introduced among SER/HI because CBT and other psychotherapies could be more prevalent in competing with SSRIs. It could be thought that we could have included a longer period for outcome measurement. However, our main interest is using SSRIs as an indicator of having a new diagnosis of CMDs and our approach is conservative in considering the occurrence of SABs close to the prescription of SSRIs since there is mixed evidence about the long-term efficacy of SSRIs.[29 48] Another limitation is that we did not have access to a level of detail of work to explore whether a person in non-standard arrangements is scheduled for work or is between work periods. Further research with alternative data sources is needed in this regard.

One of the strengths of this study is that it is based on population registers with high validity and low attrition rates, which allows for accurate and objective measurement of PE through a multidimensional measure.[49] Similarly, the linkage of pharmaceutical and work disability registers with the exact dates of SSRI prescription and SABs allowed for an accurate assessment of the outcome. The study is based on a large subpopulation of the Swedish working population. Lastly, the longitudinal study design reduces reverse causation (ie, health selection into PE).

## Generalisability

The results of this study are specific to our Swedish study population with a prescription of SSRIs and may not be generalisable to other countries with large variations in the treatments available for CMDs, subsidisation of SSRIs costs and SABs regulations. Additionally, it is important to note that this study specifically focuses on the effects of PER on the onset of CMDs, and it is possible that the onset of other health conditions, particularly those that are physically related (not subjected to stigma as mental health), may lead to different patterns of SABs.

## Concluding remarks

This study observed a trend of lower use of SABs among people with CMDs in PER and SER/HI than in SER, particularly, women with a period of three consecutive years in PER. Accordingly our findings suggest that precariously employed workers may face barriers to accessing SABs. More inclusive policies are recommended to reduce these obstacles and guarantee SABs for those who are in PER and/or reduce involuntary forms of SP.

**Author affiliations**
[1]Unit of Occupational and Environmental Medicine, Institute of Environmental Medicine, Karolinska Institutet, Stockholm, Sweden
[2]Division of Insurance Medicine, Department of Clinical Neuroscience, Karolinska Institutet, Stockholm, Sweden
[3]Department of Public Health Sciences, Stockholm University, Stockholm, Sweden
[4]Section of Epidemiology, Department of Public Health, University of Copenhagen, Copenhagen, Denmark
[5]Lawrence S. Bloomberg Faculty of Nursing, University of Toronto, Toronto, Ontario, Canada
[6]Dalla Lana School of Public Health, University of Toronto, Toronto, Ontario, Canada
[7]Department of Mental Health, The Johns Hopkins University Bloomberg School of Public Health, Baltimore, Maryland, USA
[8]MAP Centre for Urban Health Solutions, Li Ka Shing Knowledge Institute, Unity Health Toronto, Toronto, Ontario, Canada
[9]School of Nursing, Cape Breton University, Sydney, Nova Scotia, Canada
[10]ESIMar (Mar Nursing School), Parc de Salut Mar, Universitat Pompeu Fabra-affiliated, Barcelona, Spain
[11]SDHEd (Social Determinants and Health Education Research Group), IMIM (Hospital del Mar Medical Research Institute), Barcelona, Spain
[12]GREDS (Research Group on Health Inequalities, Environment, and Employment Conditions Network), Universitat Pompeu Fabra, Barcelona, Spain
[13]Centre for Occupational and Environmental Medicine, Stockholm Region, Stockholm, Sweden

**Acknowledgements** The authors are grateful to all the researchers of the PWR Consortium and acknowledge Anette Linnersjö for ordering and receiving data and the valuable input on the data management. The authors also acknowledge feedback provided by Cecilia Orellana and Fabrizio Méndez Rivero during various stages of the development of the study.

**Contributors** TB conceived the initial idea of the study and act as a guarantor of the work. All authors contributed to the design of the study. TB and JCH-R further developed the study design and discussed it with NM-S. Data management and data analysis were performed by JCH-R with support from NM-S. All authors contributed to the interpretation of the results. JCH-R drafted the first version of the manuscript, and all authors made subsequent revisions of the work, agreed on the text and findings, and approved the final version. The corresponding author certifies that all listed authors meet authorship criteria.

**Funding** This research was funded by FORTE—Swedish Research Council for Health Working Life and Welfare. Grant number 2021-00034.

**Competing interests** None declared.

**Patient and public involvement** Patients and/or the public were not involved in the design, or conduct, or reporting, or dissemination plans of this research.

**Patient consent for publication** Not applicable.

**Ethics approval** The Regional Ethics Board of Stockholm granted ethical permission for the study (no. 2017/1224-31/2 and 2018/1675-32). This is a register-based study and informed consent is not applicable.

**Provenance and peer review** Not commissioned; externally peer reviewed.

**Data availability statement** Data may be obtained from a third party and are not publicly available.

**ORCID iDs**
Julio C Hernando-Rodriguez http://orcid.org/0000-0003-0878-667X
Chantelle Murley http://orcid.org/0000-0003-4150-4275

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
