## [Reviewer comments · BMJ Open]

ARTICLE DETAILS

TITLE (PROVISIONAL)	Unequal access? Use of sickness absence benefits by precariously employed workers with common mental disorders: A register-based cohort study in Sweden
AUTHORS	Hernando-Rodriguez, Julio C.; Matilla-Santander, Nuria; Murley, Chantelle; Blindow, Katrina; Kvarn, Signild; Almroth, Melody; Kreshpaj, Bertina; Thern, Emelie; Badarin, Kathryn; Muntaner, Carles; Gunn, Virginia; Padrosa, Eva; Julià, Mireia; Bodin, Theo

VERSION 1 – REVIEW

REVIEWER	Bryce, Andrew The University of Sheffield, Economics
REVIEW RETURNED	23-Mar-2023

GENERAL COMMENTS	This paper uses administrative data from Sweden to explore the association between precarious employment and sickness absenteeism due to common mental health disorders. The study finds a U-shaped pattern in the results, with people in both precarious employment and high-income standard employment being less likely to take sickness absence than those in lower income standard employment. In the context of an evolving labour market, where non-standard forms of employment are becoming more commonplace, and increasing prevalence (or at least awareness) of mental health conditions, these are very interesting findings. Moreover, the authors make use of a rich administrative dataset which helps this paper to compare favourably with studies based on survey data. My main concern with the paper is the lack of statistically significant results across all the models tested. This substantially undermines the conclusions drawn out from the analysis. For example, in Table 2 the only significant result for the PER group is for women with three years of continuous exposure. I don't think this is enough to provide conclusive evidence that people with PER are less likely to take sickness absence. One possible solution to improve the precision of the estimates is to keep everybody in the sample (including those who did not receive SSRI treatment) and estimate an interaction term between employment relationship and SSRI. This would increase the sample size substantially and should reduce the confidence intervals. This would also allow the authors to identify whether there is any association between employment relationship and sickness absence among those not receiving SSRI treatment. The paper could also be much improved by using a panel data model. Rather than estimating the probability of SAB given
---

	employment situation in the current period, the authors could consider estimating a model that identifies changes in SAB probability when people move into and out of PER. This should reduce the potentially confounding influence of unobserved individual characteristics. I also have a few other comments:  • Another potential confounding variable is tenure in current job (other studies have shown that people who are new to their job are less likely to take SA) so the authors should consider also controlling for this if the data are available. • The comment at the bottom of page 7 suggesting that women concentrating in certain jobs is an explanation for their lower SA does not seem valid as sector and occupation are controlled for in the analysis. • Is it possible to measure severity of mental health disorder, for example by observing level of SSRI prescription? It would be interesting to see if people in PER experience more acute spells of mental ill-health than people in standard employment, as this would lead to higher propensity for SA. • There are papers in the social sciences literature that focus on a similar topic, so these should also be cited. For example, see the following references: Garcia-Mainer, I., Green, P. & Navarro Paniagua, M., (2018). The effect of permanent employment on absenteeism: Evidence from labour reform in Spain. ILR Review, 71(2):525-549. Goerke, L. & Pannenberg, M., (2015). Trade union membership and sickness absence: Evidence from a sick pay reform. Labour Economics, 33:13-25. Kahn, J. & Rehnberg, C., (2009). Perceived job security and sickness absence: A study on moral hazard. European Journal of Health Economics, 10:421-428. Mastekaasa, A., (2013). Unionization and certified sickness absence: Norwegian evidence. ILR Review, 66(1):117-141.
--	---

REVIEWER	Luan, Chu University of Saskatchewan
REVIEW RETURNED	28-Mar-2023

GENERAL COMMENTS	This is a well-written manuscript comparing the use of sickness absence benefits due to common mental disorders between precarious and non-precarious workers with CMD. There are some issues to be clarified as follows:  -The PE measure may change over time. Thus, Logistic regression models may not tackle this. In this paper, I assume the PE measure was measured at baseline. - The outcome was the first occurrence of SABs due to CMD starting in the 2 months spanning the month before and after the SSRI prescription date in 2017. The wording of 'first' makes me
---

	think of "survival analysis". Why the time-to-event was not considered in the analyses instead of logistic regression? - Why did the authors conduct stratification analyses (by sex)? Did the authors test the interaction effects (if any) of sex and the main exposure? The baseline characteristics of males and females may be different. Will the results be similar to 2 model approaches: 1) interaction effect of sex and main exposures in the model; and 2) sex-specific analyses as the authors conducted?
--	--

VERSION 1 – AUTHOR RESPONSE

Reviewer: 1

Dr. Andrew Bryce, The University of Sheffield Comments to the Author:

This paper uses administrative data from Sweden to explore the association between precarious employment and sickness absenteeism due to common mental health disorders. The study finds a U-shaped pattern in the results, with people in both precarious employment and high-income standard employment being less likely to take sickness absence than those in lower income standard employment.

In the context of an evolving labour market, where non-standard forms of employment are becoming more commonplace, and increasing prevalence (or at least awareness) of mental health conditions, these are very interesting findings. Moreover, the authors make use of a rich administrative dataset which helps this paper to compare favourably with studies based on survey data.

My main concern with the paper is the lack of statistically significant results across all the models tested. This substantially undermines the conclusions drawn out from the analysis. For example, in Table 2 the only significant result for the PER group is for women with three years of continuous exposure. I don't think this is enough to provide conclusive evidence that people with PER are less likely to take sickness absence. One possible solution to improve the precision of the estimates is to keep everybody in the sample (including those who did not receive SSRI treatment) and estimate an interaction term between employment relationship and SSRI. This would increase the sample size substantially and should reduce the confidence intervals. This would also allow the authors to identify whether there is any association between employment relationship and sickness absence among those not receiving SSRI treatment.

We appreciate the reviewer's insights and constructive feedback. We understand the concern about the lack of statistically significant results. We would like to clarify our approach to interpreting the results of the models by considering confidence intervals as "compatibility intervals" as recommended by Amrhein et al. (2019). Rather than solely focusing on statistical significance, we examined the effect sizes that are most compatible with our data. In Table 2, the overall association found between 1-year exposure to PER and SABs (adjusted OR=0.92; CI95% 0.81-1.05) means a risk difference compatible with a reduction of 19% to a small increase of 5% (a reduction of 25% to a small increase of 4% among women). We think these findings are valuable because they indicate a stronger negative association rather than an increased risk of sickness absence (values closer to the point estimate are more compatible than those in the limits). That's why we included near-significance estimates in the results and we discussed them in the study. However, we have made some revisions in the main text to be clear with our approach and we have rewritten more modest conclusions as follows:

"The study indicates that workers with CMD in precarious employment may use SABs to a lower extent. Accordingly, there is a need for (i) guaranteeing access to SABs for people in precarious employment and/or (ii) reducing involuntary forms of presenteeism." (page 2, Abstract, last

paragraph).

“This study observed a trend of lower use of SABs among people with CMD in PER and SER/Hi than in SER, particularly, women with a period of three consecutive years in PER. Accordingly our findings suggest that precariously employed workers may face barriers to accessing SABs. More inclusive policies are recommended to reduce these obstacles and guarantee SABs for those who are in PER and/or reduce involuntary forms of SP.” (page 14, concluding remarks).

“We would like to clarify our approach to interpreting the results of the models by considering confidence intervals as “compatibility intervals” as recommended by Amrhein et al. [34]. Rather than solely focusing on statistical significance, we examined the effect sizes that are most compatible with our data.” (page 6, methods, last paragraph).

“This association between PER and SABs suggests 8% reduced risk. However, our data is also compatible with a range of risk differences, spanning from a reduction of 19% to a small increase of 5%.” (page 9, results, last paragraph).

Regarding the suggestion of keeping everybody in the sample (including those without SSRI treatment), to improve the precision of the estimates, we would like to clarify our reasons for focusing exclusively on people with a new diagnosis of CMD (i.e., a new SSRI treatment). Our study specifically targets this group because we aim to investigate the use of sickness absence benefits. By including individuals who did not receive SSRI treatment, we would not be able to distinguish whether a lower risk of sickness absence among the precariously employed group is attributed to their better health or their reduced use of sickness absence benefits. Therefore, our study design focus solely on people with a new diagnosis of CMD allows us to specifically examine the relationship between precarious employment and the use of sickness absence benefits, without reiterating what has already been explored by other previous studies (Matilla, 2020; Oke, 2016).

References:

Amrhein, V., Greenland, S., & McShane, B. (2019). Scientists rise up against statistical significance. *Nature*, 567(7748), 305-307.

Matilla-Santander, N., González-Marrón, A., Martín-Sánchez, J. C., Lidón-Moyano, C., Cartanyà-Hueso, À., & Martínez-Sánchez, J. M. (2020). Precarious employment and health-related outcomes in the European Union: a cross-sectional study. *Critical Public Health*, 30(4), 429-440.

Oke, A., Braithwaite, P., & Antai, D. I. D. D. Y. (2016). Sickness absence and precarious employment: a comparative cross-national study of Denmark, Finland, Sweden, and Norway. *The international journal of occupational and environmental medicine*, 7(3), 125.

The paper could also be much improved by using a panel data model. Rather than estimating the probability of SAB given employment situation in the current period, the authors could consider estimating a model that identifies changes in SAB probability when people move into and out of PER. This should reduce the potentially confounding influence of unobserved individual characteristics. To address potential unobserved confounding problems in our study, we opted not to use a panel data model approach. We think that this approach could introduce similar unobserved confounding problems as our current study design. Instead, we employed a Directed Acyclic Graph (DAG) to identify and adjust for all potential confounders measured in our data. Furthermore, since we measure the exposure and outcome with minimal follow-up time, we are confident this helps to minimize the problems associated with residual varying-confounding (changes over time).

I also have a few other comments:

- Another potential confounding variable is tenure in current job (other studies have shown that people who are new to their job are less likely to take SA) so the authors should consider also controlling for this if the data are available.

Thank you for pointing this out. The multidimensional measurement of precarious employment already accounts for tenure in the current job with the item of “temporariness” (stable employment: having the same employer for three years; unstable: having the same employer for less than 3 years as referred in Jonsson, 2021).

Reference:

Jonsson J, Matilla-Santander N, Kreshpaj B, et al. Exploring multidimensional operationalizations of precarious employment in Swedish register data – a typological approach and a summative score approach. *Scand J Work Environ Health* 2021;47:117–26. doi:10.5271/sjweh.3928

- The comment at the bottom of page 7 suggesting that women concentrating in certain jobs is an explanation for their lower SA does not seem valid as sector and occupation are controlled for in the analysis.

Thank you for raising this point. We have carefully considered your feedback and revised the text accordingly. We have nuanced that part of the text by adding the following sentence:

“However, this explanation is unlikely to apply to our study since we accounted for the potential effect of women’s concentration in certain sectors on SABs by controlling for the sector in the analysis.” (page 12, discussion, last paragraph).

Furthermore, we would like to mention that although we included both, sector, and occupation as covariates, we ultimately excluded occupation in the models as confounder. This decision was based on the observation that including occupation in the models barely changed the estimates (as mentioned on page 9, results, end of the paragraph).

- Is it possible to measure severity of mental health disorder, for example by observing level of SSRI prescription? It would be interesting to see if people in PER experience more acute spells of mental ill-health than people in standard employment, as this would lead to higher propensity for SA.

Thank you for pointing this out. Although it is possible to measure the severity of mental ill-health and investigate if people in PER experienced more acute spells of mental ill-health compared to those in SER, our study focused on investigating changes in the use of sickness absence benefits among people with a new diagnosis of CMD (a new SSRI treatment). We excluded individuals with severe and recurrent episodes of CMD, as individuals who are seriously ill are more likely to use sickness absence benefits regardless of whether they have low or high employment quality (Aaviksoo, 2016; Ziebarth, 2013).

References:

Aaviksoo, E., & Kiivet, R. A. (2016). Influence of the sickness benefit reform on sickness absence. *Health Policy*, 120(9), 1070-1078.

Ziebarth, N. R. (2013). Long-term absenteeism and moral hazard—Evidence from a natural experiment. *Labour Economics*, 24, 277-292.

- There are papers in the social sciences literature that focus on a similar topic, so these should also be cited. For example, see the following references:

Garcia-Mainer, I., Green, P. & Navarro Paniagua, M., (2018). The effect of permanent employment on absenteeism: Evidence from labour reform in Spain. *ILR Review*, 71(2):525-549.

Goerke, L. & Pannenberg, M., (2015). Trade union membership and sickness absence: Evidence from a sick pay reform. *Labour Economics*, 33:13-25.

Kahn, J. & Rehnberg, C., (2009). Perceived job security and sickness absence: A study on moral hazard. *European Journal of Health Economics*, 10:421-428.

Mastekaasa, A., (2013). Unionization and certified sickness absence: Norwegian evidence. *ILR Review*, 66(1):117-141.

Thank you for drawing our attention to these articles. We have incorporated several of the recommended references (Goerke, 2015; Mastekaasa, 2013) into our manuscript. These references provide additional insights that complement the cited systematic review on risk factors of SA

(Allebeck, 2004), which does not extensively cover studies on unionization and sickness absence. Furthermore, we have included a systematic review about temporary employment and health (including sickness absence) (Virtanen, 2005) and a scoping review about determinants of SA and RTW among employees with CMD (de Vries, 2018).

As a result of these additions, we have revised a section of the introduction to reflect these updated references as follows:

“Previous studies have examined demographic characteristics such as age and gender, singular employment characteristics, trade union membership or work environment factors in relation to SA [12–15]. Some of these determinants of SA have been examined specifically among people with common mental disorders (CMD – stress-related disorders, depression, and anxiety) [16]. To our knowledge, only two previous cross-sectional studies have analysed the relationship between PE and SA, and both of which found a positive association [17,18]. This is in line with previous studies confirming a relationship between PE and poor health, including a systematic review of longitudinal studies [19]. A recent Swedish study using the same data sources as the present study found that trajectories with constant PE increased the risk of CMD [20].”

References:

Allebeck P, Mastekaasa A. Chapter 5. Risk factors for sick leave - general studies. *Scand J Public Health* 2004;32:49–108. doi:10/dvr35c

Virtanen, M., Kivimäki, M., Joensuu, M., Virtanen, P., Elovainio, M., & Vahtera, J. (2005). Temporary employment and health: a review. *International journal of epidemiology*, 34(3), 610-622.

de Vries, H., Fishta, A., Weikert, B., Rodriguez Sanchez, A., & Wegewitz, U. (2018). Determinants of sickness absence and return to work among employees with common mental disorders: a scoping review. *Journal of occupational rehabilitation*, 28, 393-417.

Reviewer: 2

Dr. Chu Luan, University of Saskatchewan Comments to the Author:

This is a well-written manuscript comparing the use of sickness absence benefits due to common mental disorders between precarious and non-precarious workers with CMD.

There are some issues to be clarified as follows:

-The PE measure may change over time. Thus, Logistic regression models may not tackle this. In this paper, I assume the PE measure was measured at baseline.

Thank you for your insights and constructive feedback. We appreciate your comment on the importance of the changing exposure to precarious employment and model suitability. Yes, precarious employment is measured at baseline in 2016. In this study, we primarily focus on assessing whether people access to sickness absence benefits based on their recent employment status, with prior employment status or changes being of secondary importance. As part of the sensitivity analyses, rather than the main analyses, we also measured the exposure in 2014-2016. The sensitivity analysis focused on examining the effect of stability on the outcome by considering three consecutive years (2014-2016) with the same exposure. Given the short follow-up duration in our study, we think that using logistic regression models aligns well with our research approach.

- The outcome was the first occurrence of SABs due to CMD starting in the 2 months spanning the month before and after the SSRI prescription date in 2017. The wording of 'first' makes me think of "survival analysis". Why the time-to-event was not considered in the analyses instead of logistic regression?

Thank you for bringing up this question. As we have a shorter follow-up period, we think the time factor is less important, and the logistic regression is well suited. Also, our study interest is not how long it takes before sickness absence benefits are granted but rather whether one is or is not granted sickness absence benefits during the follow-up period, which is suitable for logistic regression. Future studies with aims further exploring the association identified in the current study are required, and rightly, could use Cox regressions.

- Why did the authors conduct stratification analyses (by sex)? Did the authors test the interaction effects (if any) of sex and the main exposure? The baseline characteristics of males and females may be different. Will the results be similar to 2 model approaches: 1) interaction effect of sex and main exposures in the model; and 2) sex-specific analyses as the authors conducted?

Thank you for raising this comment. We conducted the analyses stratified by gender to account for potential differences in sickness absence behaviours as shown in previous studies (mentioned on page 6, statistical analysis).

Also, we followed your suggestion and examined the interaction effects between gender and employment quality. We observed that women in precarious employment had a lower risk of sickness absence benefits compared to men in standard employment after controlling for the same potential confounding factors (the adjusted odds ratio of the main effect and the interaction was 0.73). We should note that the effect of the interaction term was somewhat pronounced for women in PER, but, in general, differences between genders and employment quality categories were relatively small.

However, when comparing the model without the interaction term to the model containing it, the likelihood ratio test was not statistically significant ($p\text{-value} > 0.3955$). This suggests that including the interaction term does not improve the fit of the model. Therefore, we have kept the original model without the interaction term in the manuscript as it adequately captures the relationship between employment quality, gender, and sickness absence benefits based on the current data.

We have modified the manuscript to incorporate the interaction analysis as follows:

“Interaction effects of gender and the main exposure were tested.” (page 6, statistical analysis).

“We tested interaction effects between gender and employment quality. Women in precarious employment had 27% lower risk of sickness absence benefits compared to men in standard employment after controlling for the same potential confounding factors (data not shown). The effect of the interaction term was somewhat pronounced for women in PER, but, in general, differences between gender and employment quality categories were relatively small. However, when comparing the model without the interaction term to the model containing it, the likelihood ratio test was not statistically significant ($p\text{-value} > 0.3955$) (data not shown). This suggests that including the interaction term does not improve the fit of the model. Therefore, we have kept the original model without the interaction term.” (page 10, results).

Reviewer: 1

Competing interests of Reviewer: None

Reviewer: 2

Competing interests of Reviewer: No

VERSION 2 – REVIEW

REVIEWER	Bryce, Andrew The University of Sheffield, Economics
REVIEW RETURNED	02-Jun-2023

GENERAL COMMENTS	I appreciate that the authors have done all they can to control for potential confounders. However, it is still the case that selection into different types of employment is non-random, so there remains the possibility of unobserved factors being correlated both with the probability of PER and the probability of SABs. Hence, I think this should be stated as a limitation of the study design when trying to infer a causal impact of employment quality on SABs. With regards to generalisability, it might also be emphasised that the current study is concerned only with the effects of an onset of CMD. It is possible that the onset of other health conditions (including those not related to mental health) could lead to very
---

	different patterns of sickness absenteeism and presenteeism. Due to the stigmatised nature of mental health, there is evidence to suggest that mental ill-health may be regarded by both employees and employers as a less acceptable reason for SA than physical ill-health, and could drive higher levels of presenteeism. Relevant papers include Johns and Xie (1998) and Dewa and Lin (2000), although both are quite dated now. Dewa, C.S. & Lin, E., (2000). Chronic physical illness, psychiatric disorder and disability in the workplace. Social Science & Medicine, 51:41-50. Johns, G. & Xie, J.L., (1998). Perceptions of absence from work: People's Republic of China versus Canada. Journal of Applied Psychology, 83(4):515-560.
--	--

REVIEWER	Luan, Chu University of Saskatchewan
REVIEW RETURNED	09-Jun-2023

GENERAL COMMENTS	The authors' responses are adequate and I do not have any follow-up questions.
--

VERSION 2 – AUTHOR RESPONSE

Reviewer: 1

Dr. Andrew Bryce, The University of Sheffield

Comments to the Author:

I appreciate that the authors have done all they can to control for potential confounders. However, it is still the case that selection into different types of employment is non-random, so there remains the possibility of unobserved factors being correlated both with the probability of PER and the probability of SABs. Hence, I think this should be stated as a limitation of the study design when trying to infer a causal impact of employment quality on SABs.

We appreciate the reviewer's comment regarding the limitations of our study design. We acknowledge the possibility of residual confounding and its potential impact on inferring a causal relationship between employment quality and SABs. To address this concern, we have included a specific statement in the limitations section of the manuscript as follows:

"Another limitation is the possibility of residual confounding when inferring the impact of employment quality on SABs." (Page 17, limitations and strengths section).

With regards to generalisability, it might also be emphasised that the current study is concerned only with the effects of an onset of CMD. It is possible that the onset of other health conditions (including those not related to mental health) could lead to very different patterns of sickness absenteeism and presenteeism. Due to the stigmatised nature of mental health, there is evidence to suggest that mental ill-health may be regarded by both employees and employers as a less acceptable reason for SA than physical ill-health, and could drive higher levels of presenteeism. Relevant papers include Johns and Xie (1998) and Dewa and Lin (2000), although both are quite dated now.

Dewa, C.S. & Lin, E., (2000). Chronic physical illness, psychiatric disorder and disability in the workplace. *Social Science & Medicine*, 51:41-50.

Johns, G. & Xie, J.L., (1998). Perceptions of absence from work: People's Republic of China versus Canada. *Journal of Applied Psychology*, 83(4):515-560.

We appreciate the reviewer's valuable input, and we have taken his suggestion into account to

address the issue of generalizability and the potential influence of mental health stigma on sickness presenteeism in our discussion of the results. We have made the following additions to the text: “Furthermore, sickness presenteeism can be partially explained by the social stigma surrounding mental health, which may originate from colleagues and employers (47). This stigma can lead individuals to feel more validated in taking sickness absence when their illness is physical, rather than when it is related to mental health.” (Page 12, discussion section, first paragraph).

“Additionally, it is important to note that this study specifically focuses on the effects of PER on the onset of CMDs, and it is possible that the onset of other health conditions, particularly those that are physically related (not subjected to stigma as mental health), may lead to different patterns of SABs”. (Page 14, Generalizability).

Reviewer: 2

Dr. Chu Luan, University of Saskatchewan

Comments to the Author:

The authors' responses are adequate and I do not have any follow-up questions.

Thank you for your positive feedback.

Reviewer: 1

Competing interests of Reviewer: None

Reviewer: 2

Competing interests of Reviewer: No